# Schistosoma mansoni Larval Extracellular Vesicle protein 1 (SmLEV1) is an immunogenic antigen found in EVs released from pre-acetabular glands of invading cercariae

Thomas A. Gasan[1], Marije E. Kuipers[2], Grisial H. Roberts[1], Gilda Padalino[1], Josephine E. Forde-Thomas[1], Shona Wilson[3], Jakub Wawrzyniak[3], Edridah M. Tukahebwa[4], Karl F. Hoffmann[1], Iain W. Chalmers[1] *

1 Institute of Biological, Environmental & Rural Sciences (IBERS), Aberystwyth University, Edward Llwyd Building, Aberystwyth, United Kingdom, 2 Department of Parasitology, Leiden University Medical Centre, Leiden, Netherlands, 3 University of Cambridge, Department of Pathology, Tennis Court Road, Cambridge, United Kingdom, 4 Vector Control Division, Ugandan Ministry of Health, Kampala, Uganda

* iwc@aber.ac.uk

## Abstract

Extracellular Vesicles (EVs) are an integral component of cellular/organismal communication and have been found in the excreted/secreted (ES) products of both protozoan and metazoan parasites. Within the blood fluke schistosomes, EVs have been isolated from egg, schistosomula, and adult lifecycle stages. However, the role(s) that EVs have in shaping aspects of parasite biology and/or manipulating host interactions is poorly defined. Herein, we characterise the most abundant EV-enriched protein in Schistosoma mansoni tissue-migrating schistosomula (Schistosoma mansoni Larval Extracellular Vesicle protein 1 (SmLEV1)). Comparative sequence analysis demonstrates that lev1 orthologs are found in all published Schistosoma genomes, yet homologs are not found outside of the Schistosomatidae. Lifecycle expression analyses collectively reveal that smlev1 transcription peaks in cercariae, is male biased in adults, and is processed by alternative splicing in intra-mammalian lifecycle stages. Immunohistochemistry of cercariae using a polyclonal anti-recombinant SmLEV1 antiserum localises this protein to the pre-acetabular gland, with some disperse localisation to the surface of the parasite. S. mansoni—infected Ugandan fishermen exhibit a strong $IgG_1$ response against SmLEV1 (dropping significantly after praziquantel treatment), with 11% of the cohort exhibiting an IgE response and minimal levels of detectable antigen-specific $IgG_4$. Furthermore, mice vaccinated with rSmLEV1 show a slightly reduced parasite burden upon challenge infection and significantly reduced granuloma volumes, compared with control animals. Collectively, these results describe SmLEV1 as a Schistosomatidae-specific, EV-enriched immunogen. Further investigations are now necessary to uncover the full extent of SmLEV1's role in shaping schistosome EV function and definitive host relationships.

**Data Availability Statement:** All relevant data are within the manuscript and its Supporting Information files.

**Funding:** TAG was funded by an IBERS, Aberystwyth University PhD studentship. IWC and KFH were funded by the European Union's Seventh Framework Programme (FP7/2007-2013; http://ec. europa.eu/research/fp7/index_en.cfm) under grant agreement number 242107 and IWC has funded partly by the Higher Education Funding Council for Wales (HEFCW) - Global Challenges Research Fund (GCRF) https://www.hefcw.ac.uk/en/ publications/circulars/w20-16he-global-challenges-research-fund-2020_21/. The funders had no role in study design, data collection and analysis, decision to publish, or preparation of the manuscript.

**Competing interests:** The authors have declared that no competing interests exist.

## Author summary

From the moment they start to penetrate the human skin, juvenile schistosomes secrete bioactive molecules to facilitate invasion, evade the host immune response, and establish a long-lived infection. One component of these secretions are pre-packaged, membrane-bound extracellular vesicles (EVs) capable of modulating the host immune response. Whilst EVs contain a subset of known biologically relevant molecules, many EV proteins remain uncharacterised. Here we investigate the most abundant protein found within EVs released by early skin-stage schistosomes—SmLEV1. We show that the *lev1* gene is only found in the schistosomes and closely related species and is most abundantly expressed during the early stages of human infection. Localising the SmLEV1 protein to the pre-acetabular glands of the cercarial head, highlights this gland as a source of larval EVs. Vaccination with SmLEV1 had minimal effects on parasite burden, but significantly reduced the liver granuloma volume, implicating SmLEV1 as a potential anti-pathology target for schistosomiasis. Our collective findings are the first characterisation of SmLEV1 and provide essential data and resources for the ongoing study of schistosome EVs.

## Introduction

Human schistosomiasis is caused by infection with platyhelminth species of the genus *Schistosoma*. Three main species are infectious to humans and include *Schistosoma mansoni*, *Schistosoma haematobium*, and *Schistosoma japonicum* [1]. Predominantly found in parts of Asia, South America, and Africa, but also recently found in Corsica [2], schistosomiasis is the most important macroparasitic neglected tropical disease (NTD) in terms of its global health impact [3]. Schistosome infection is initiated when cercariae (free-living, non-feeding, aquatic larvae) penetrate human skin and transform into tissue migrating schistosomula. Concomitant with skin penetration, schistosomula release many excreted/secreted (ES) products to facilitate invasion and modulate the host immune response [4–6]. Membrane-bound EVs represent a component of schistosomula ES products and are enriched in proteins as well as small non-coding RNAs [7].

It is clear that EVs are produced by many other species of complex multicellular helminths such as *Heligmosoides polygyrus*, *Fasciola hepatica*, *Echinostoma caproni* (reviewed in [8]) as well as other *S. mansoni* lifecycle stages [7,9–11]. In addition to containing a number of bioactive molecules, *S. mansoni* EVs have the capacity to modulate the response of host immune cells. For example, *S. mansoni* schistosomula EVs can be internalised by human monocyte-derived dendritic cells (moDC) via glycan-mediated binding to DC-SIGN leading to an increase in expression of co-stimulatory molecules (CD80, CD86), the regulatory surface marker PD-L1, and an increased expression of IL-12 and IL-10 [12]. In addition, EVs from adult *S. mansoni* worms can be directly internalised by T-helper (Th) cells *in vitro*, downregulating Th2 T-cell differentiation independently of host antigen presenting cells (APC) [13]. Microvesicles and exosome-like vesicles derived from adult worms are also internalised by human umbilical vein endothelial cells (HUVECs), via tetraspanin-mediated internalisation, wherein they downregulate multiple genes associated with immune cell regulation, proliferation, and cell signalling [14].

Overall, evidence suggests that schistosome EVs can modulate host immune responses. However, in many cases, the responsible bioactive molecule(s) have not been elucidated. Therefore, in-depth functional characterisation of schistosome EV cargo components could reveal further details of reported immunomodulatory properties or potential host interactions.

Towards this goal, we herein characterise the most abundant protein component of EVs derived from mechanically transformed schistosomula, *S. mansoni* Larval Extracellular Vesicle-1 (SmLEV1; smp_074560).

## Results

### *smlev1* is unique to Schistosomatidae and exhibits cassette-based alternative splicing

The full-length SmLEV1 protein sequence (smp_074560.1) contains no identifiable Pfam domains [15], nor signal peptide or transmembrane domains (identified using SignalP 3.0 [16] or TMHMM v 2.0 [17] respectively). BLAST searches of related and representative species identified *smlev1* homologues in all *Schistosoma* species with a published genome, and in the closely related species *Trichobilharzia regenti* (S1 Fig and S1 Table.). However, no homologues were identified in any other parasitic or non-parasitic platyhelminth species. When searches were broadened to representative host species (e.g., *Homo sapiens*), *smlev1* homologues were also not found. Initial PCR amplification of full-length *smlev1* (starting ATG to stop codon) from cercaria cDNA resulted in three products visualised by DNA gel electrophoresis (Fig 1). Subsequent DNA Sanger sequencing of these products identified 4 splice variants of *smlev1*, termed *smlev1.1—smlev1.4*. Further amplification and sequencing of 7-week adult male worm

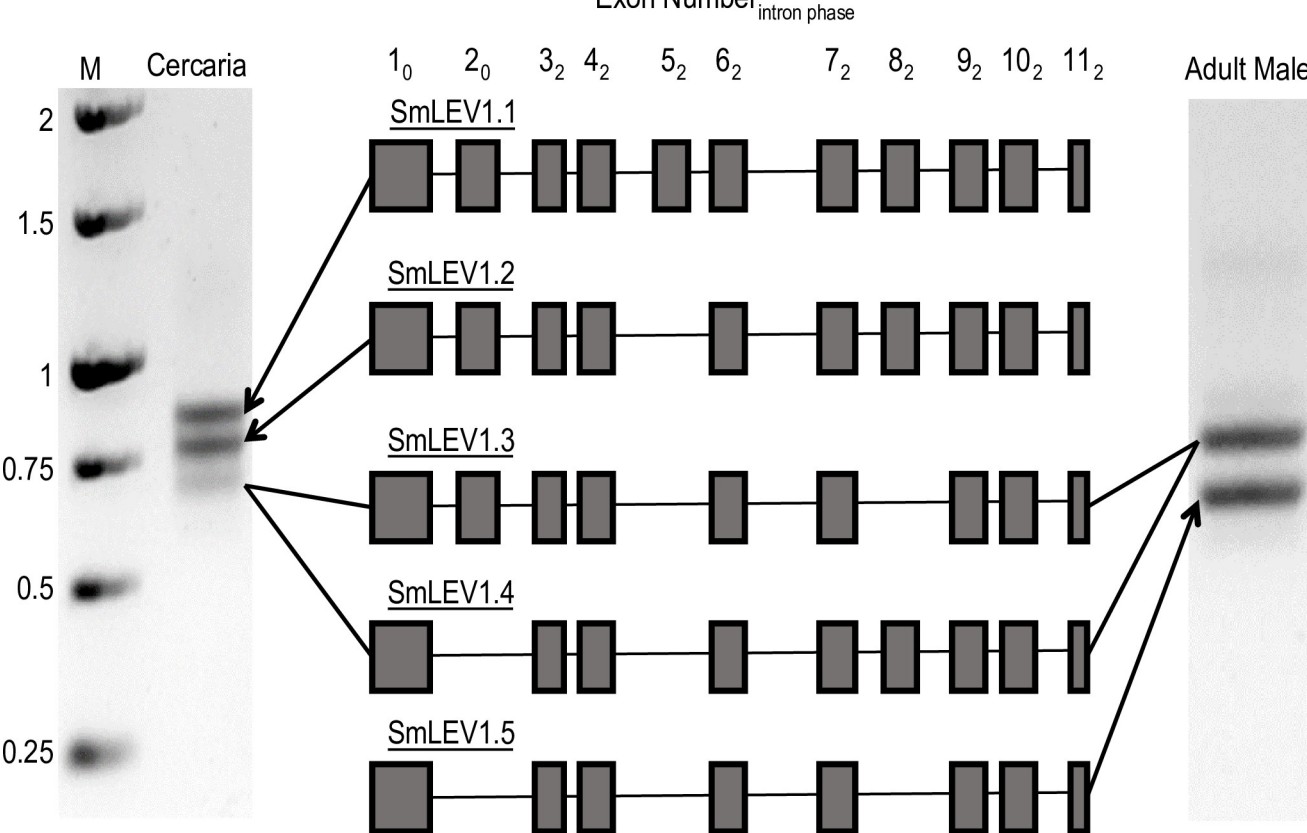

**Fig 1. *smlev1* exhibits cassette-based alternate splicing.** PCR amplification of *smlev1* from cercarial and adult worm cDNA revealed the full-length, genome prediction (*smlev1.1*) in addition to four shorter splice variants (*smlev1.2—smlev1.5*). *smlev1.1* (full length) = 891 bp, *smlev1.2* = 810 bp, *smlev1.3* = 726 bp, *smlev1.4* = 702 bp, *smlev1.5* = 618 bp. Shaded boxes represent exons present in the transcript; the exon number and subscript 3′ intron phase of each exon is indicated in the rubric. M = 1 Kbp ladder, sizes in Kilo base pairs.

cDNA revealed the presence of both *smlev1.3* and *smlev1.4*, alongside an additional splice variant—*smlev1.5* (Fig 1). Gene model visualisation and multiple sequence alignment showed that the *smlev1* isoforms exhibited cassette-based alternative splicing, with *smlev1.1* containing all 11 exons, *smlev1.2* missing exon 5, *smlev1.3* missing exon 5 and 8, *smlev1.4* missing exons 2 and 5, and finally *smlev1.5* missing exons 2, 5, and 8 (Fig 1). Due to the exon phases of *smlev1*, the reading frame remained intact in these alternative splice variants and the amino acids encoded in downstream exons were not impacted.

## *smlev1* expression is developmentally regulated

Analysis of previous *S. mansoni* RNA-seq data [18], showed a significantly higher transcript abundance of *smlev1* in cercariae compared to 3 h schistosomula (p<0.05), 24 h schistosomula (p<0.01), male (p<0.05) and female (p<0.01) adult worms, as well as a 15.5-fold male-biased expression in the adult worm (p<0.05) (Fig 2A). Analysis of previously available DNA microarray data [19] supports the significant cercarial *smlev1* abundance and a 10.7-fold male biased expression in the dioecious adults (S2A Fig). The *S. japonicum lev1* homologue (Sjp_0090520), termed *sjlev1*, exhibited a similar pattern of expression to *smlev1*, (derived from analysis of a previous DNA microarray study [20]) with abundant cercarial and adult male-biased expression (S2B Fig).

## *smlev1* exon usage is varied across the parasite lifecycle

qRT-PCR studies of *smlev1* were performed to confirm the life-cycle profile and assess whether the alternative splicing of exons 2, 5 and 8 was developmentally regulated. Specific qRT-PCR amplification of exon 1 (containing the starting methionine and found on all isoforms) and the variable *smlev1* exons (exons 2, 5 and 8) revealed a significantly different picture of exon usage in cercariae, compared to the schistosomula and adult worm stages (Fig 2B). The quantification of all *smlev1* isoforms, as determined by qRT-PCR amplification of exon 1, closely resembled the cercarial-dominant expression pattern observed in the RNA-seq (Fig 2A) and DNA microarray data (S2A Fig). However, the variable exon 2 was present in approximately 77% of cercarial *smlev1* transcripts; this was reduced to 50% at 24 h after transformation to schistosomula (p<0.05) and further reduced to 30% at 72 h post transformation (p<0.01). Similarly, exon 5 was present in 27% of cercarial *smlev1* transcripts, whilst only 7–11% of *smlev1* transcripts from schistosomula or adult worms contained this exon (p<0.005). Exon 8 was present in 43–51% of cercarial and schistosomula *smlev1* transcripts, with no significant variation in abundance detected for these lifecycle stages. However, the presence of exon 8 in *smlev1* transcripts was significantly higher in adult worms compared to cercariae (p<0.01 male and p<0.05 female) with 80–85% of the total adult worm *smlev1* transcripts containing this exon. Interestingly, the percentage abundance of exon 2 or exon 8 was not significantly different between male and female worms. However, the abundance of exon 5 was significantly lower in transcripts from female worms (8%) compared to male worms (10%, p<0.05).

## SmLEV1 is found in a distinct population of schistosomula EVs

Recombinant, His$_6$-tagged SmLEV1.3 (approx. MW 29 kDa) was produced in NiCo21 (DE3) *E. coli* [21] and purified by Ni-agarose (NA) chromatography. SDS-PAGE separation and subsequent mass spectrometric analysis revealed N-terminal breakdown during production of rSmLEV1.3, resulting in three protein bands (S3 Fig). Each of these bands contained only rSmLEV1.3 peptides; no *E. coli* proteins or other SmLEV1 isoforms were present (S2 Table). Pure rSmLEV1.3 (comprised of all three proteins) was used to produce anti-rSmLEV1.3 polyclonal antibody (pAb) in rabbits.

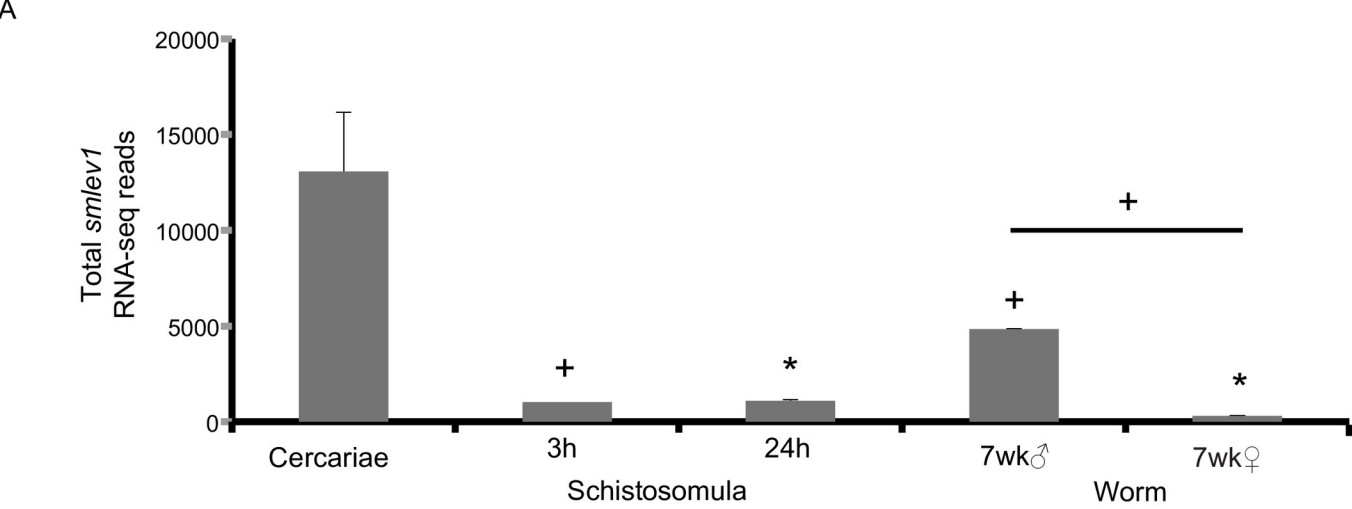

**Fig 2. *smlev1* is differentially spliced across the schistosome lifecycle.** (A) The *smlev1* transcript is enriched in cercariae, validating previous RNA-seq data, and shows a 15.5-fold male-bias expression in the adult worm [18]. The total *smlev1* reads is shown on the y axis and lifecycle stage on the x axis; error bars represent standard deviation. (B) Abundance of three variable *smlev1* exons, represented as a % of total transcript. Each cluster of columns reflects one variable exon (shown in red bold numbering) with individual columns for each lifecycle stage, shown in the legend. In both figures, significance (determined by ANOVA and post-hoc T-test analysis) is expressed as $p < 0.05$ +, $p < 0.01$ *, or $p < 0.005$ **; $p < 0.001$ ***.

Using the anti-SmLEV1.3 pAb to probe soluble extracts from various life stages revealed immunoreactive bands ~24 kDa in mass, in each of the larval stages (cercariae, 3 h and 72 h schistosomula) but not in the adult worm, or egg stages (Fig 3A, orange box). Two major immunoreactive bands approximately ~28–35 kDa in mass, (Fig 3A, red box) and two minor reactive bands ~70–80 kDa in mass (Fig 3A, blue arrows) were additionally detected in 3 h schistosomula. An immunoreactive band ~45 kDa was detected in 3 h schistosomula however, this band was also reactive with the rabbit pre-bled serum (Fig 3A, green star) and is therefore not specific to the anti-SmLEV1.3 pAb.

EVs from the culture supernatant of 0–72 h schistosomula were isolated using identical methods to those described previously [12]. Probing these EVs with the anti-SmLEV1.3 pAb resulted in a somewhat similar pattern to 3 h schistosomula, with two prominent immunoreactive proteins of approximately 28 kDa and 35 kDa in mass detected (Fig 3B, red box). These two proteins likely represent SmLEV1.1 (34 kDa) and SmLEV1.3 (28 kDa), previously identified in the 0–72 h schistosomula EV proteome [7]. Two ~65 kDa proteins were also identified in the schistosomula EV proteome (Fig 3B, blue arrow) as well as the non-specific ~45 kDa protein (Fig 3B, green star) however, the ~24 kDa proteins were not detected in EVs. EVs from adult worms (AW) did not contain any immunoreactive proteins when probed with anti-SmLEV1 pAb (Fig 3B).

Based on these findings, EVs from early stage schistosomula (0–72 h) (obtained as before) were subjected to further iodixanol gradient fractionation (fractions ranging from 1.076 to 1.28 g/cm$^3$) and probed with the anti-SmLEV1.3 pAb. These experiments revealed major reactivity in fractions 5–7 (1.2 g/cm$^3$ to 1.16 g/cm$^3$) with minor reactivity in fraction 1 and 4 (1.28 g/cm$^3$ and 1.23 g/cm$^3$ respectively) (Fig 3C). Similar to the total schistosomula EV sample (Fig 3B), all EV fractions contained two major reactive proteins at approximately 28–35 kDa (Fig 3C, red box). In addition, fractions 5–7 showed a double-doublet of larger reactive proteins (around 55 to 65 kDa, blue arrows), again similar to what was observed in Fig 3B. In the highly reactive fractions (fractions 5–7), the non-specific ~45 kDa protein was also detected (Fig 3C, green star). Importantly, no proteins in the EV-depleted supernatant (sup) exhibited anti-SmLEV1.3 reactivity (Fig 3C). In attempt to further characterise the SmLEV1-containing EVs according to the minimal information for studies of extracellular vesicles (MISEV) guidelines [22], the same 0–72 h schistosomula EV fractions were also probed with an Ab that recognises *S. mansoni* tetraspanin 2 (SmTSP-2) [9,23] (Fig 3D). Here, multiple bands were visualised, likely reflecting the mono/dimeric nature of SmTSP-2 as discussed previously [9]. Immunoreactivity was predominantly found in schistosomula EV fractions 6 and 7 (1.182–1.158 g/cm$^3$), which we show is consistent with the adult worm EV fractions containing SmTSP-2 reactivity (1.26–1.06 g/cm$^3$) (S4 Fig) also seen in previous investigations [9].

## SmLEV1 is localised to the pre-acetabular glands in cercariae

As transcriptome and WB analyses of schistosome samples indicated that *smlev1*/SmLEV1 was most abundantly found in cercariae/early schistosomula (and EVs derived from these lifecycle stages), immunohistochemistry (IHC) was conducted on cercariae utilising the anti-SmLEV1.3 pAb (Fig 4). In addition, fluorescein labelled peanut agglutinin (PNA) was used to stain the complete acetabular gland system as described previously [24]. Co-staining was not possible, therefore, cercariae were stained with either anti-SmLEV1.3, rabbit pre-bleed antisera (control) or PNA. SmLEV1 predominantly localised to a region consistent with the pre-acetabular glands (Fig 4A, indicated by white arrows) as determined by comparative staining with PNA (Fig 4B, white arrows). As expected PNA also stained the post-acetabular glands (Fig 4B, yellow arrows) as well as anterior acetabular ducts (Fig 4B, purple arrows). These regions were

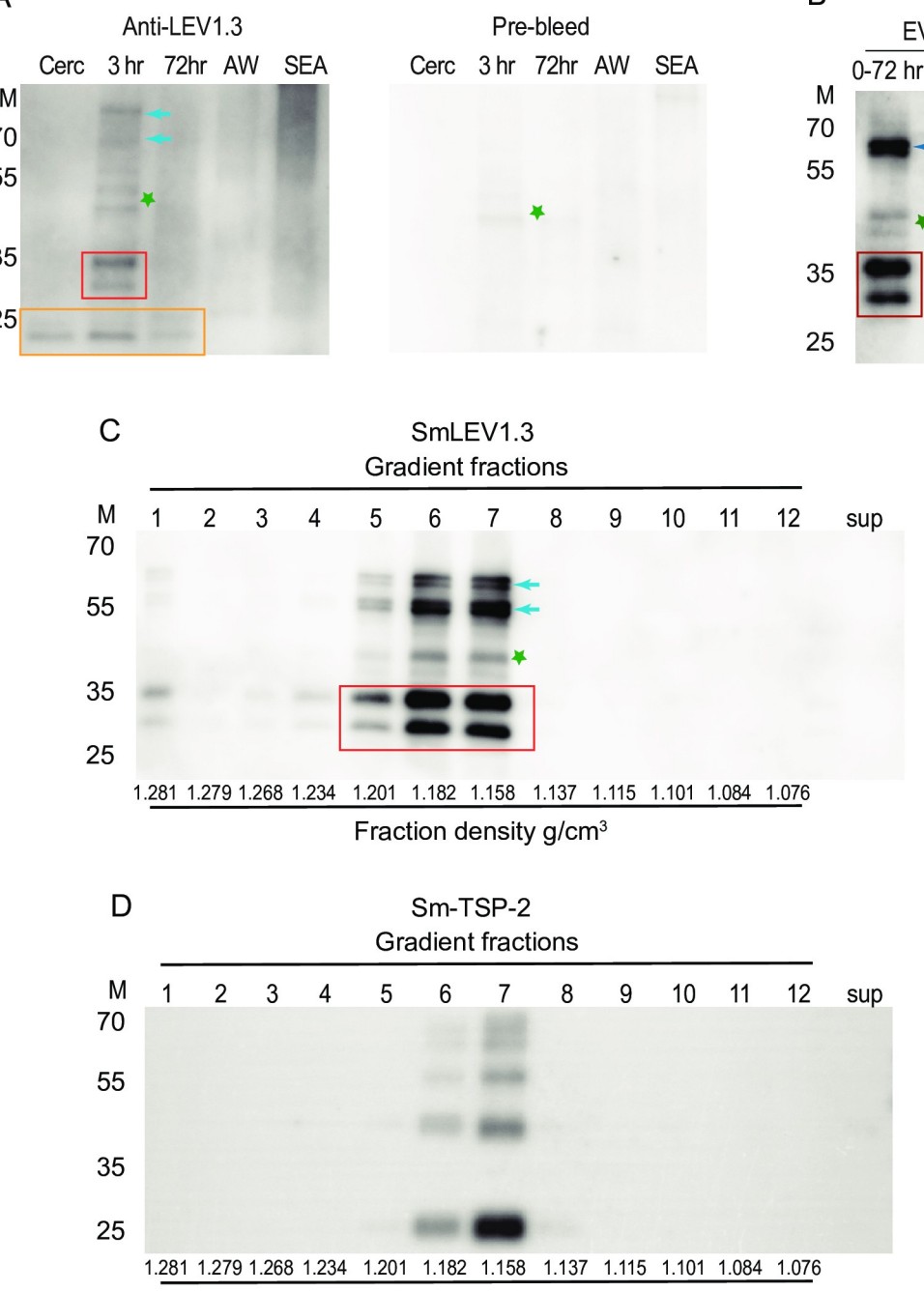

**Fig 3. SmLEV1 is enriched in schistosomula EVs with densities spanning 1.16 to 1.2 g/cm³.** (A) Parasite extracts (1.5 μg) from different life stages were probed with anti-SmLEV1.3 or rabbit pre-bleeds (primary Ab 1:2,000). Cerc–Cercariae, 3 h– 3 hour schistosomula, 72 h– 72 hour schistosomula, AW–mixed sex adult worm, SEA–Soluble egg antigen. (B) Anti-SmLEV1.3 probing EVs (100 ng) from 0–72 h schistosomula (0–72 h) or mixed-sex adult worms (AW). (C) EVs obtained from 0–72 h cultivated schistosomula were separated via density gradient prior to anti-SmLEV1.3 or (D) anti-SmTSP-2 probing. Fraction numbers are indicated above the blot; the respective densities in g/cm³ are shown below the plot. In all figures, the red box indicates the major 28–35 kDa proteins, blue arrows indicate additional larger proteins, the orange box indicates slightly smaller ~24 kDa reactive proteins. Green star indicates a non-specific immunoreactive protein. For probing EVs, primary pAb was used at 1:20,000 dilution. For all figures, secondary anti-rabbit HRP conjugated antibody was used at 1:10,000. M = protein standard, mass in kilo Daltons.

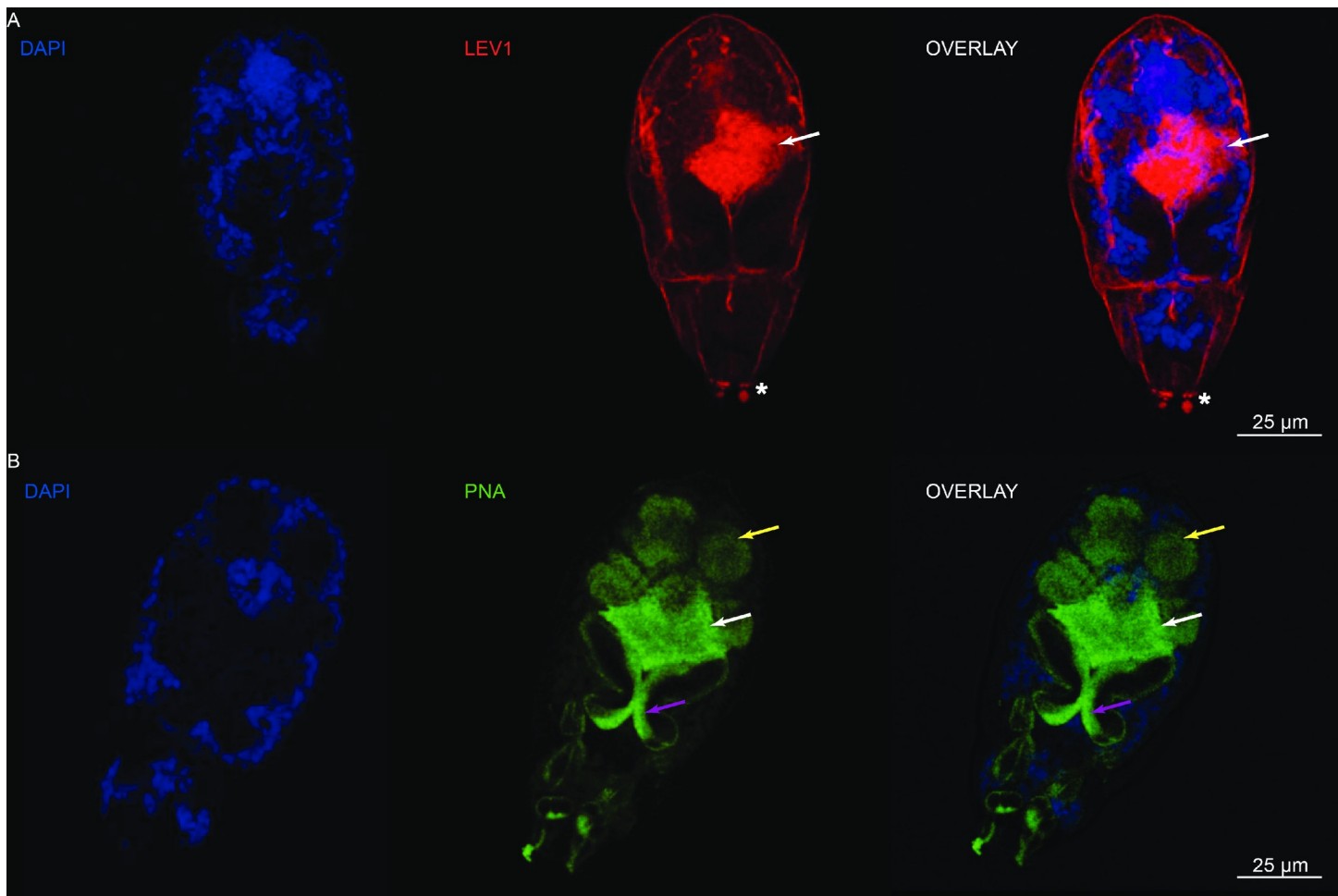

**Fig 4. SmLEV1 is concentrated in cercarial pre-acetabular glands.** (A) Permeabilised cercariae were stained with anti-SmLEV1.3 pAb (1:50) followed by goat anti-rabbit Alexa Fluor 594 secondary antibody (1,500). (B) Alternatively, permeabilised cercariae were stained with fluorescein labelled PNA (4 μg/ml). Subsequently, all stained cercariae were stained with DAPI (1 μg/ml) before mounting and visualisation. White arrows indicate the staining of the pre-acetabular gland with anti-SmLEV1.3 pAb (A) and PNA (B). Yellow arrows indicate post-acetabular glands and purple arrows indicate anterior acetabular ducts stained with PNA. Asterisk indicates staining of peripheral extensions also stained by rabbit pre-bleeds (S5 Fig).

not stained with anti-SmLEV1.3, implying SmLEV1 is restricted to the pre-acetabular portion of this secretion system. Some disperse SmLEV1 signal was also seen on the surface of the cercariae (Fig 4A). Anti-SmLEV1.3 pAb reactivity against the anterior end peripheral extensions (Fig 4A; indicated by asterisk), was also found in cercariae samples stained with the rabbit pre-bleed control (S5 Fig); therefore, this staining was not specific.

## Schistosome-infected humans recognise rSmLEV1.3

As SmLEV1 appears to be enriched in EVs derived from infective stage cercariae/tissue migrating schistosomula, human plasma antibody reactivity was subsequently measured in a cohort of *S. mansoni* infected Ugandan males (aged 7–50 years) from a high transmission area [25] (Fig 5). Plasma was taken at two time-points, before and 9 weeks after treatment with praziquantel (PZQ). SmLEV1.3 "responders" were classified as those that exhibited a pre-treatment antibody titre greater than the mean + 3 standard deviations of the response by the non-endemic samples. To elucidate the age profiles of the responders, pre-treatment anti-rSmLEV1.3 IgG$_1$, IgG$_4$, and

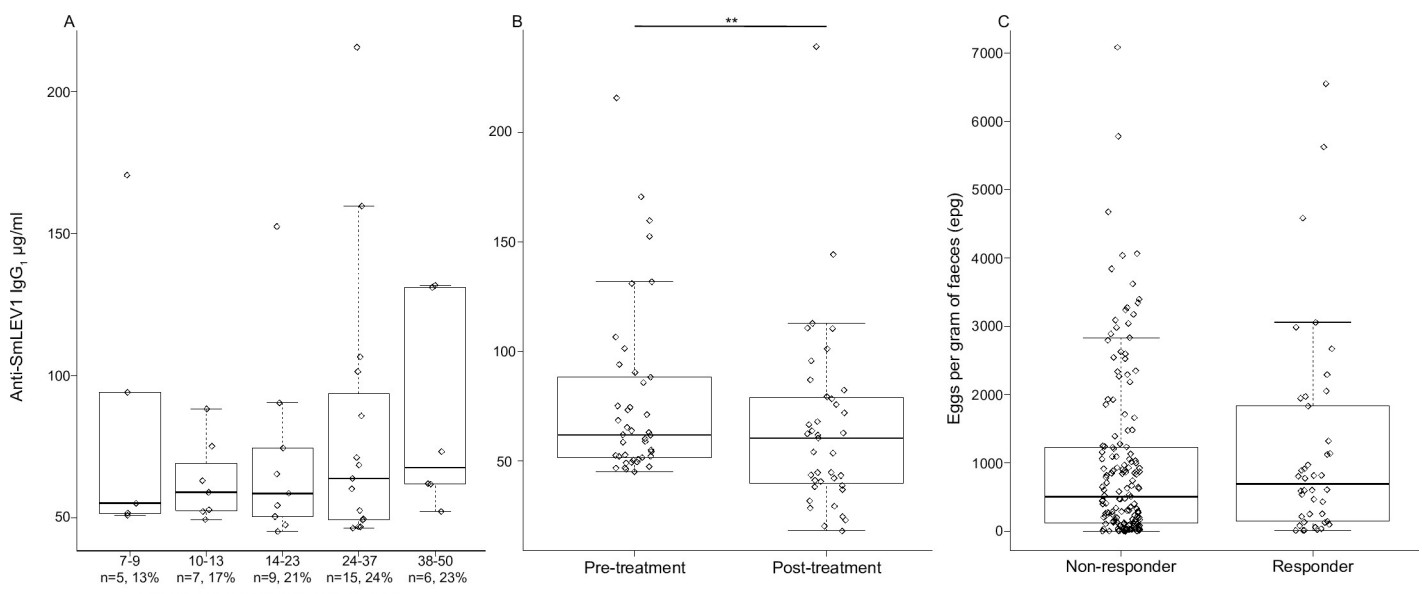

**Fig 5. Anti-SmLEV1 IgG$_1$ responses drop after PZQ treatment in an endemic human cohort.** Descriptions of cohort selection, quantitative parasitology, and treatment regimens for this study can be found in a previous publication [25]. (A) The level of anti-SmLEV1.3 IgG$_1$ in responders does not significantly change across the different age groups. The number and percentage of responders are shown for each age group. (B) The level of anti-SmLEV1.3 IgG$_1$ drops after treatment with praziquantel (p<0.005, **). (C) The egg burden of SmLEV1.3 responders and non-responders does not significantly differ. The boxplots show the median (line) with the upper and lower quartile (box) as well as the upper and lower observation within 1.5x interquartile range (IQR) (whiskers). Individual data points are also plotted.

IgE levels were plotted for five age groups (7–9, 10–13, 14–23, 24–37, 38–50). Before PZQ treatment, a detectable IgG$_1$ response to rSmLEV1.3 was present in 21% of the cohort. While SmLEV1.3 responders were not statistically associated with any one age group (p = 0.738) (Fig 5A), there was a statistically significant drop in the amount of anti-SmLEV1.3 IgG$_1$ antibodies after treatment with PZQ (p<0.005) across the responding cohort (42 individuals) (Fig 5B). Responders to rSmLEV1.3 contained approximately 20% more eggs per gram of faeces, although this was not statistically significant (*p* = 0.74) (Fig 5C). Only one individual (0.5%) exhibited a detectable IgG$_4$ response before treatment and only one different individual after treatment with PZQ. Before PZQ treatment, 11% of individuals exhibited an IgE response to rSmLEV1.3, which did not significantly correlate with age (S6A Fig). The number of IgE responders did not significantly decrease after treatment (S6B Fig).

## Vaccination with SmLEV1.3 significantly reduced granuloma volume in the murine model of schistosomiasis

Having established that SmLEV1 could induce an immunogenic response in an endemic human cohort, we next wanted to assess the potential of rSmLEV1.3 as a protective immunogen in an experimental model of schistosomiasis (Fig 6). Here, C57BL/6 mice were immunised three times, with either the adjuvant formulation containing 20 μg recombinant rSmLEV1.3 (n = 14) or the adjuvant formulation alone (n = 14). All mice were subsequently challenged 35 days after the final vaccination and sacrificed 7 weeks later. Blood was taken from all mice 1 day prior to the first vaccination, 7 days after the final vaccination, 7 days after cercarial challenge, and termination (Fig 6A). Vaccinated mice showed a significant increase in anti-SmLEV1.3 IgG$_1$ levels after vaccination (p<0.005) and a further increase 1 week after parasite challenge (p<0.005). A significant decrease in anti-SmLEV1.3 IgG$_1$ levels was seen 7 weeks

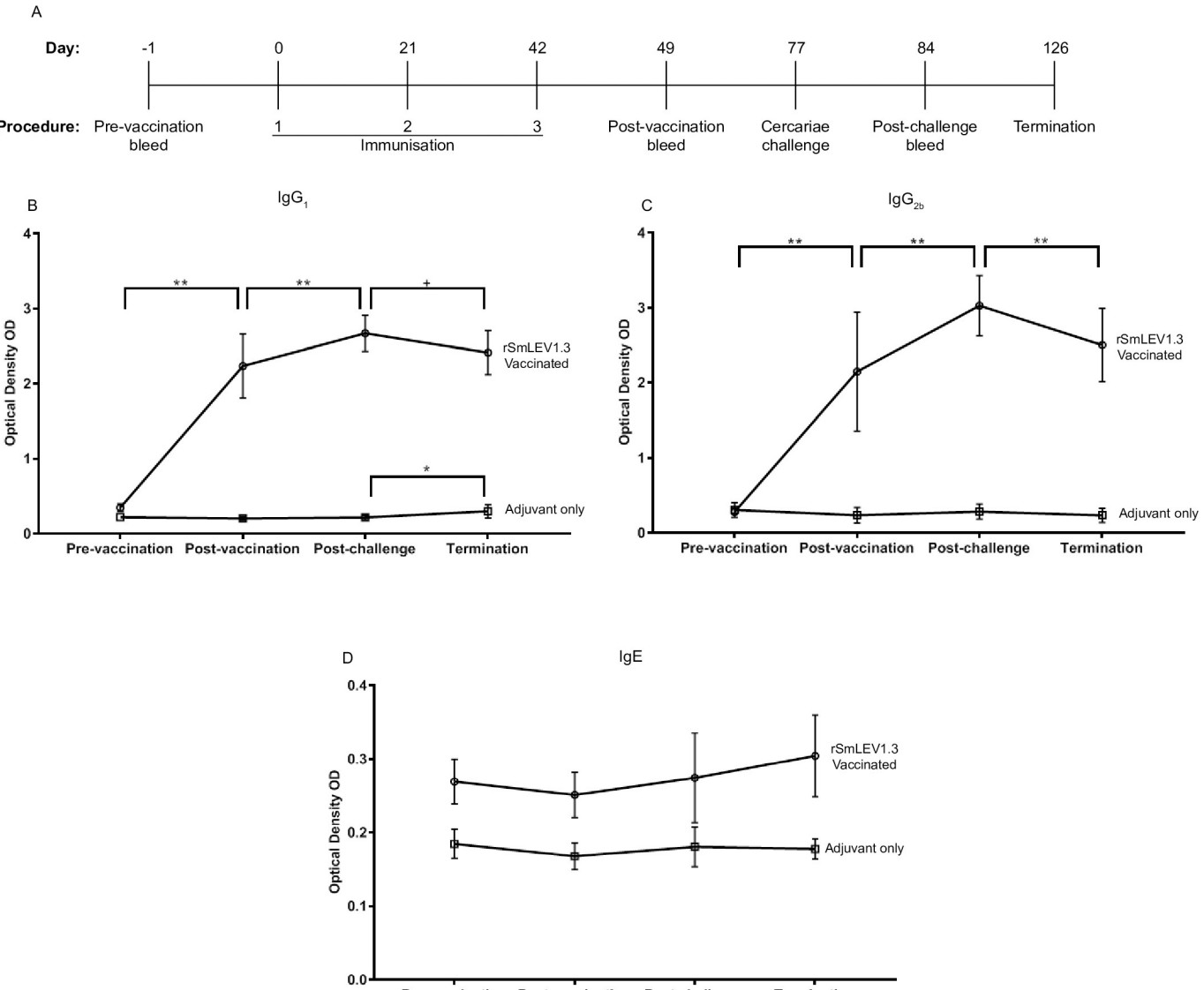

**Fig 6. rSmLEV1.3 vaccination induces anti-SmLEV1 IgG$_1$ and IgG$_{2b}$ responses, which are further increased 7 days after cercarial challenge.** (A) SmLEV1 vaccine regime, showing the 3 immunisations given 21 days apart and the times of each bleed. All mice were challenged with 150 cercariae, 35 days after final immunisations. (B-D) At each timepoint, serum levels of anti-rSmLEV1.3 specific IgG$_1$, IgG$_{2b}$, and IgE were measured by ELISA, and represented as optical density (OD). With all figures, the line and whiskers represent the mean with the standard deviation. Significance is expressed as +(p<0.05), *(p<0.01) or **(p<0.005). rSmLEV1.3 vaccine (circles, n = 14) or adjuvant alone (squares, n = 14).

post–infection compared to after parasite challenge (p<0.05). Control adjuvant-only mice showed a significant increase in anti-SmLEV1.3 IgG$_1$ antibodies only upon termination 7 weeks after parasite challenge (p<0.01) (Fig 6B). Overall, the pattern of anti-SmLEV1.3 IgG$_{2b}$ levels during the vaccination regime reflected that seen for IgG$_1$, with a significant increase after vaccination and parasite challenge (p<0.005) and a significant drop upon termination, compared to post-challenge. However, no significant increase in SmLEV1 IgG$_{2b}$ antibodies was seen in the control animals between post-challenge and termination (Fig 6C). No significant difference was seen in the level of anti-SmLEV1.3 IgE between any time points in vaccinated or control animals (Fig 6D).

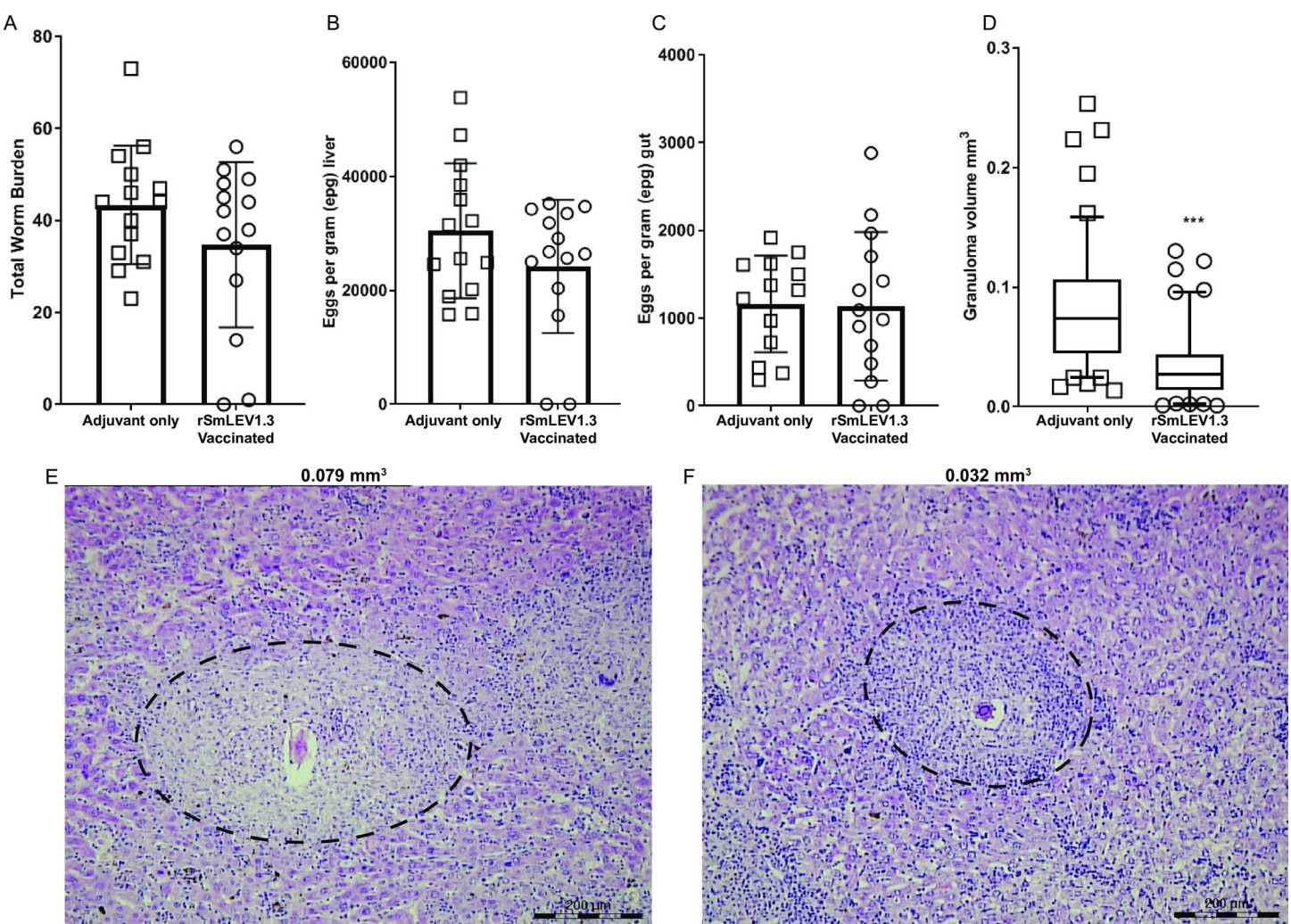

**Fig 7. rSmLEV1.3 vaccination led to a non-significant reduction in adult worm and egg burdens, but a significant reduction in egg-induced liver granuloma volumes.** (A) Upon termination of the vaccine regime (7 weeks post cercarial challenge) the mean number of adult worms was 17% lower in vaccinated (n = 14) animals compared to control animals (n = 14). (B) In addition, rSmLEV1 vaccination led to a 21% reduction in the number of liver-resident eggs. (C) The number of gut-resident eggs was similar between treatment groups. (D) rSmLEV1.3 vaccinated mice showed a greater than 60% reduction in liver granuloma volumes (p<0.001, ***). The boxplot shows the median (line) with the upper and lower quartile (box) as well as the 5–95 percentile (whiskers). Each data point is the average granuloma volume per mouse, from 10 individual granulomas. Histological sections of liver were taken from control (E) and vaccinated (F) mice and stained with H&E. Granulomas with volumes closest to the mean for each treatment group were chosen for illustration.

Upon perfusion and when compared to control adjuvant-only animals, mice vaccinated with rSmLEV1.3 contained fewer adult worms (41 and 34 respectively, Fig 7A) as well as fewer liver-resident eggs (30,470 mean epg liver and 24,170 mean epg liver respectively, Fig 7B) and intestinal-entrapped eggs (1,162 mean epg gut and 1,148 mean epg gut respectively, Fig 7C). However, no reduction was statistically significant. Worms also did not differ in size (as measured by area) between vaccinated and control animals (S7 Fig). Importantly, SmLEV1.3 vaccinated mice exhibited significantly smaller egg-induced liver granulomas (mean volume = 0.033 mm$^3$) compared to control mice (mean volume = 0.086 mm$^3$) (Fig 7D). This vaccine-related effect represented a 62% reduction in granuloma volume (p<0.001). Representative images of H&E-stained liver sections were chosen to illustrate the average granuloma size for control (Fig 7E, 0.079 mm$^3$) and vaccinated animals (Fig 7F, 0.032 mm$^3$).

## Discussion

The long-term survival of schistosomes in the definitive host depends on effective host immune modulation. As recent evidence demonstrates that EVs derived from skin-stage schistosomula [12] and adult worms [13,14] regulate cellular responses of surrogate host models, these vehicles of parasite-to-host communication are likely essential elements of this survival process. Detailed characterisation of schistosome EV components could reveal molecular clues to tease apart how schistosome EVs specifically contribute to host modulation. Here, we have focused our investigation on the *Schistosoma mansoni* Larval Extracellular Vesicle -1 (SmLEV1) protein, the most abundant protein component of mechanically transformed schistosomula EVs (derived from 0–72 h *in vitro* cultures) [7].

We found *lev1* homologues across the *Schistosoma* genus, in addition to the closely related avian trematode *T. regenti* (S1 Fig) but no homologues were identified in any other parasitic or free-living genera. Therefore, it appears that *lev1* is specific to the *Schistosomatidae*, with no homologues outside of this family. Both *smlev1* and *sjlev1* homologues [20] exhibit cercarial-dominant expression (S2 Fig) with qRT-PCR data additionally showing differential alternative splicing of *smlev1* in this lifecycle stage when compared to schistosomula and adult worms (Fig 2). Alternative splicing of the three variable *smlev1* exons is responsible for at least 5 variants of SmLEV1, ranging in size from 24.2–34.7 kDa. In conjunction with transcript abundance, alternative splicing could be employed to control SmLEV1's downstream function, as seen in other *S. mansoni* proteins [26,27].

In-line with the larval-dominant *smlev1* transcript abundance, we detected anti-SmLEV1 reactive proteins in total extracts from cercariae, 3 hour, and 72 hour schistosomula (Fig 3A), with the highest abundance in the 3 h schistosomula. This supports a previous observation of SmLEV1 in the proteins released by mechanically transformed cercariae, cultured for 3 h (0–3 h RP) (found in supplementary data of [28]). A 24 kDa reactive protein, consistent in size with SmLEV1.5 (24. 2 kDa) was found in all larval-stage extracts, whilst the 3 h schistosomula also contained reactive proteins consistent with SmLEV1.1 (34.7 kDa), and another shorter isoform–either SmLEV1.2 = 31.7 kDa, SmLEV1.3 = 28.3 kDa, or SmLEV1.4 27.23 kDa. The identity of the larger immunoreactive proteins (55–70 kDa) is currently unknown, however this size range is consistent with dimerisation of the SmLEV1 isoforms. Our data also confirms the presence of SmLEV1 in EVs derived from the supernatant of schistosomula cultured from transformation to 72 h (0–72 h) (Fig 3B) as shown previously [7]. Furthermore, our data reinforces this association by showing that anti-SmLEV1.3 reactivity was enriched in the same density fractions as Sm-TSP-2, a typical marker for *S. mansoni* EVs [9] (Fig 3C and 3D).

Interestingly, anti-SmLEV1.3 immunoreactivity was much lower in the endpoint 72 h schistosomula extract when compared with the 3 h schistosomula proteome (Fig 3B). *smlev1* transcript abundance is highest in cercariae, with dramatic reduction at 3 h post-transformation and beyond (as seen by RNA-seq (Fig 2A) and DNA microarray (S2 Fig)), with recent single cell RNA-Seq (scRNA-Seq) analysis of 2-day schistosomula additionally confirming this observation [29]. Given our data, we contend that cercariae and/or newly transformed schistosomula package SmLEV1 into EVs that are subsequently excreted/secreted into the culture supernatant where it remains. By 72 h post-transformation, transcription diminishes and all SmLEV1 protein has been transported out of the parasite. This timeframe fits with the activity of the cercarial acetabular glands, which secrete their contents during the cercarial and early schistosomula stages but are largely emptied by 72 h post-transformation [30]. The acetabular glands consist of three pairs of post-acetabular glands, two pairs of pre-acetabular glands, and the associated ducts leading to the anterior of the parasite [31,32]. Mucous-like secretions from the post-acetabular glands aid in skin adhesion [33], whilst pre-acetabular glands secrete

substances to facilitate skin penetration, such as proteinases [34] and calcium-binding proteins [35] to mediate calcium-based post-translational regulation of proteinase activity [36]. Supporting our contention of EV packaging in the acetabular glands, we show that a SmLEV1.3 pAb was reactive towards a region of the cercarial head, consistent with the pre-acetabular gland (Fig 4)—as well as some disperse staining on the surface of the parasite. It has been previously hypothesised that the acetabular secretion system is a source of cercarial/schistosomula EVs [7]; our data now provides evidence for SmLEV1-containing vesicles originating in the pre-acetabular gland, alongside products with known involvement in skin penetration.

Similar to other secreted antigens [23,37], we hypothesised that schistosome-infected individuals would exhibit an anti-SmLEV1 antibody response. The chosen cohort to test this hypothesis was chronically infected male individuals from a schistosomiasis-endemic fishing community in Uganda, formerly described by Fitzsimmons *et al.* [25]. This cohort has been previously investigated for serological responses to other *S. mansoni* antigens, including the surface bound SmLy6B [38], and the cryptic, sub-surface antigen SmTAL1 [25]. Over 20% of this cohort were seropositive for SmLEV1.3-IgG$_1$, but IgG$_1$ response was not associated with any age group (Fig 5A). Interestingly the significant drop in SmLEV1.3-IgG$_1$ after PZQ treatment (Fig 5B) suggests that some SmLEV1 may be produced by adult worms. Any adult worm-derived SmLEV1 would likely originate from the tegument, as the *smlev1* transcript was recently localised to adult worm tegumental cells [39], which fits our observation of disperse SmLEV1 staining on the surface of the cercariae (Fig 4). However, our western blot analysis did not detect SmLEV1 in adult worms (soluble proteomes or EVs) nor in eggs (soluble proteomes) (Fig 3A). Moreover, no SmLEV1 isoforms or homologues were identified in previous proteomic studies of adult worm ES [40], adult worm EVs [9–11,41], adult worm tegument [42], and egg ES [43,44]. Therefore, it seems most likely that the abundance of SmLEV1 expressed by adult worms and /or eggs is below the level of sensitivity of our anti-SmLEV1 pAb or standard proteomic analysis.

In the human cohort IgG$_4$ response to SmLEV1 was negligible, and only 11% of individuals were seropositive for anti-SmLEV1-IgE. This cohort also responded to SmLy6B in a similar fashion; wherein 25% were seropositive for Ly6B-IgG$_1$, the Ly6B-IgG$_1$ response did not correlate with age, and antibody levels dropped after treatment [38]. By comparison, the response of this cohort to SmTAL1 is markedly different, with nearly half of all individuals exhibiting an IgG$_4$ or IgE response (45% in either instance) and only 12% seropositive for anti-SmTAL1-IgG$_1$ [25]. In addition, the response to SmTAL1 shows an age-related build-up and significant increase in antibody levels after treatment, as this sub-surface antigen is exposed to the immune system upon natural- or drug mediated-parasite death. The dominant IgG$_4$/IgE and age-related response is typical of other cryptic *S. mansoni* antigens, such as the gut associated Cathepsin B1 [45]. Both SmLEV1 and SmLy6B, however, are non-cryptic antigens and elicit a dominant IgG$_1$ response in-line with the response to other EV antigens such as the aforementioned Sm-TSP-2 [23] and Glyceraldehyde 3-Phosphate Dehydrogenase [37] both of which has been identified as potential schistosomiasis vaccine candidates [46,47].

Due to the dominant IgG$_1$ response (and minor IgE response), a vaccine trial was conducted in mice to examine whether immunisation with SmLEV1.3 would confer protection upon parasite challenge. To prevent exaggerated reductions in parasite burden from non-specific pulmonary inflammation (as described by Wilson *et al* [48]), an incubation period of 35 days was chosen between the final SmLEV1.3 vaccination and parasite challenge. Vaccinated mice showed a robust serological response to SmLEV1.3 vaccination (compared to adjuvant-only animals), which was boosted upon parasite challenge. Interestingly, levels of anti-SmLEV1.3 IgG$_1$ and IgG$_{2b}$ significantly dropped at 7-weeks post-infection, likely due to a reduction in native antigen exposed to the host upon maturation of the parasites from the

single round of infection (as inferred by RNA-seq/DNA microarray data (Figs 2A and S2)) and the undetectable levels of SmLEV1 protein in adult worms (Fig 3B).

Compared to adjuvant-only animals, SmLEV1.3 vaccinated mice showed a 17% reduction in worm-burden and a 21% reduction in egg burden in the liver (Fig 7A and 7B). However, the number of eggs lodged in the gut tissue was only 1.2% lower in vaccinated animals (Fig 7C). In addition to the slight decrease in parasite burden, mice vaccinated with SmLEV1.3 exhibited granulomas less than half the size of those seen in control mice given adjuvant alone (Fig 7D, 7E, and 7F). Newly deposited, immature eggs do not induce granuloma formation [49], therefore, it is possible that the SmLEV1.3 post-immunisation response has an effect on the maturation of the tissue-entrapped egg, resulting in less secreted SEA and smaller granulomas. However, the mechanism behind this is unclear, especially considering the low *smlev1* transcript abundance in the egg and miracidia stages (S2 Fig) and undetectable protein in adult worms and eggs (Fig 3). To fully discern the anti-fecundity effect of SmLEV1.3 vaccination, more detailed egg viability and hatching assays need to be conducted, as shown with other schistosomiasis vaccine candidates [50,51].

This study provides the first detailed examination of the previously uncharacterised SmLEV1. Localisation of native SmLEV1 protein to the pre-acetabular secretion system and the cercarial surface (possibly the tegument) provides robust evidence of EVs originating from the acetabular gland of the larval stages of the parasite. Moreover, our pilot vaccination trial shows an effect of SmLEV1 vaccination on the granulomatous response to eggs. A reduced granulomatous response, concurrent with reduced worm burden, has been noted in vaccine trials with other schistosome EV proteins [52]. Therefore, targeting this pre-packaged subset of bioactive parasite molecules could be a method of ameliorating disease pathology. However, research into *S. mansoni* EVs is still young with only a few studies investigating the overall biological contents and host-modulatory activities of intact vesicles on host modulatory responses. So far, minimal functional investigations of individual components have been reported. Therefore, the characterisation of SmLEV1 herein can be used as a template for replicating detailed characterisations of other EV protein components in an attempt to understand parasite-EV-host relationships during human schistosomiasis.

## Materials and methods

### Ethics statement

All procedures performed on mice and rats adhered to the United Kingdom Home Office Animals (Scientific Procedures) Act of 1986, under the PPL 40/3092, and were approved by Aberystwyth University's Ethical Review Panel. Ethical clearance for the Musoli cohort was obtained from the Uganda National Council of Science and Technology (ethics committee for Vector Control Division, Ugandan Ministry of Health). Consent forms were translated into the local language and informed written consent was obtained from all adults and from the parents/legal guardians of all children under 15.

### Acquisition of parasite-derived material

The Puerto Rican strain (NMRI) of *S. mansoni* was perpetuated by routine passage through female, HsdOla:MF1 mice (Harlan, United Kingdom) and *Biomphalaria glabrata* (NMRI albino and pigmented hybrid) snails [53]. Mixed-sex cercariae, mechanically transformed schistosomula and adult worms were obtained as described previously [54] and either stored in Trizol at -80˚C until required for RNA extraction or immediately processed for native protein extraction (as described below). For immunohistochemistry, cercariae were fixed and permeabilised as described previously [24].

Native parasite protein extracts were prepared from PBS-washed cercariae, eggs (SEA), or 7-week adult worms (AW), using manual homogenisation, sonication and subsequent freeze/ thawing at -80˚C followed by filter sterilisation, as described previously [55]. Extracts from PBS-washed 3 h or 72 h schistosomula were prepared by a similar method, using a bead homo-geniser (MP Biomedicals), with 5 x 10 s shaking at (7.5 m/s) followed by sonication, freeze/ thawing, and filter sterilisation. Protein concentration of parasite extract was determined using BCA according to the manufacturer's protocol (Pierce BCA Protein assay kit, Thermo-Fisher Scientific).

EVs derived from 150,000 mechanically transformed schistosomula were isolated from 0–72 h *in vitro* cultures as described previously [12]. The resulting EV-enriched pellet from cultured schistosomula was either used directly or subjected to density gradient. For density gradients, the EV-enriched pellet was resuspended in 73 μL PBS with 0.2% BSA (made from overnight ultra-centrifuged PBS/5% BSA; 28,000 rpm (k-factor 265) in a SW41 Ti rotor in an Optima LE80K (Beckman Coulter)) and loaded on top of an iodixanol gradient (Optiprep, Axis Shield) consisting of 1 layer of 440 μL 60% v/v iodixanol, 1 layer of 220 μL 40% v/v iodixa-nol, 1 layer of 220 μL 30% v/v iodixanol, and 1 layer of 792 μL 10% v/v iodixanol in a thin-wall polypropylene tube for a TLS-55 rotor. The gradient was centrifuged for 2 h in an Optima TLX centrifuge (Beckman Coulter) at 50,000 rpm (average 166,180× g, k-factor 60.3, 4˚C) with slow acceleration and deceleration. Twelve gradient fractions of 145 μL were collected from the top and densities were measured using a refractometer.

Adult worm EVs were isolated by differential centrifugation and density gradient, as described previously [9] with some changes. Briefly, mixed-sex adult worms were cultured at approx. 10 worms/mL in 30 to 40 mL M199 medium (Gibco) supplemented with ABAM (Sigma) and HEPES in a 75 cm³ flask for 48 h. Culture supernatant was processed through multiple centrifugation steps (2x 10 min 200× g, 2x 10 min 500× g, 2x10 min 5,000× g) fol-lowed by high-speed centrifugation of 28,000 rpm (~100,000× g) in SW41 Ti rotor in an Optima XE (Beckman Coulter) for 65 min at 4˚C. The EV-enriched pellet was resuspended, transferred to a TLS-55 tube, and an iodixanol gradient was built on top as described above. As shown previously [9,11], fractions containing EVs (within 1.09–1.2 g/cm³) were tested for the presence of SmTSP-2 (S4 Fig) and combined for use in western blot analysis.

## Transcriptional profiling

The transcription pattern of *smlev1*, across 14 life-cycle stages, was retrieved from accessing a previous *S. mansoni* long-oligonucleotide DNA microarray study [19]. A full set of raw and normalised data is available via Array Express under the experimental accession number E-MEXP-2094. Similarly, RNA-Seq data was mined from a previous *S. mansoni* transcriptome study [56]. Significant differences in transcript abundance across the lifecycle was determined by ANOVA, followed by a Student's *t*-test post-hoc (least significant difference). Transcrip-tional profiling of the *S. japonicum* homologue (Sjp_0090520) was also achieved by accessing a previous DNA microarray study [20].

## Cloning and sequence analysis

*S. mansoni* total RNA was isolated from various lifecycle stages and 1 μg used as a template for cDNA synthesis using reverse transcription as described previously [57]. PCR primers (For-ward: 5´- ATGCCACGCTGTCGCAAAG—3´, Reverse: 5´- ATCCGTATATCTGTTATAT GT- 3´) were designed to amplify the full coding sequence of all *smlev1* isoforms using Phusion proofreading polymerase (Finnzymes). Amplicons were then cloned into the pGEMT-easy vector (Promega) and sequenced by dideoxy chain-termination sequencing (IBERS DNA

sequencing facility, Aberystwyth). Isoform sequences were visualised using Artemis [58] and multiple sequence alignments were conducted with MUSCLE [59].

## Quantitative reverse transcription (qRT)- PCR analysis

For qRT-PCR investigations, cDNA was acquired from cercariae, 24 h schistosomula, 3-day schistosomula, 7-week male- and female worms. Using the full coding sequences of the different *smlev1* isoforms, intron-spanning primers were designed to amplify constant and variable regions of the *smlev1* sequence (S3 Table). The amplicons of each primer set were sequenced in-house to ensure on-target amplification; in addition, a melt curve analysis was conducted to determine amplification specificity. All qRT-PCR amplifications were performed in triplicate with SensiFAST SYBR Hi-ROX mix (Bioline) using a StepOnePlus Thermocycler (Applied Biosystems). The amplification efficiency (E) of each primer pair was calculated by the equation $E = 10^{(-1/slope)}$ [60]. To investigate the change in abundance of each *smlev1* exon, the cycle threshold (Ct) was used in following equation to calculate a normalised expression ratio for each exon target [61]:

$$ratio = (E_{ref})^{Ct\ ref} / (E_{target})^{Ct\ target}$$

Significant difference in amplicon abundance was determined by ANOVA, followed by a Student's *t*-test post-hoc analysis (least significant difference). Data from the previous *S. mansoni* DNA microarray [19] was mined to identify a suitable reference gene across the 5 lifecycle stages investigated. The contig AF521086.1, mapping to Smp_306860 exhibited lower variation when compared to the previously used *S. mansoni* alpha tubulin (SmAT, Smp_090120) [62] (S8 Fig). Therefore, Smp_306860 was used as the reference in this study.

## SmLEV1.3 recombinant protein expression

Oligonucleotide primers were designed to incorporate *Xba*I and *Xho*I restriction sites to the respective 5´- and 3´- ends of the SmLEV1.3 isoform open reading frame sequence (Forward 5´-GTCTAGAATGCCACGCTGTCGC-3´, Reverse 5´-CATATAACAGATATACGGATCT GAGC-3´) using Phusion proofreading polymerase (Finnzymes). After PCR amplification, the product was ligated into the *Xho*I and *Xba*I digested pET30a expression vector (Novagen); this modified vector was designed to incorporate an in-frame C-terminal His$_6$ tag attached to recombinantly expressed SmLEV1.3. Before use, the SmLEV1.3/pET30a construct was sequenced in-house to confirm the correct reading frame and sequence.

SmLEV1.3/pET30a plasmids were transformed in chemically competent NiCo21(DE3) *Escherichia coli* (NEB), as per the manufacturer's instructions [21]. SmLEV1.3 expression was induced with isopropyl β-D-1-thiogalactopyranoside (IPTG) (0.1mM final concentration) and allowed to continue for 4 h at 30˚C, after which bacteria were pelleted, resuspended in binding buffer (50 mM NaH$_2$PO$_4$, 300 mM NaCl, Imidazole 10 mM) containing cOmplete Protease Inhibitor Cocktail (Roche) and lysed using a Cell Disruption System (Constant Systems). The resulting soluble and insoluble fractions were separated by centrifugation at 21,000 g for 20 min. Purification of rSmLEV1.3 was performed at 4˚C using Amintra Ni-NTA agarose beads (Expedeon) according to the manufacturer's instructions, with two additional washes. Binding buffer was used to wash the beads with increasing amounts of imidazole: 20 mM in the first wash, 75 mM in the second wash, and 150 mM for elution. The resulting protein mixture then underwent further purification with chitin-resin (NEB) to sequester the co-eluting, chitin-binding *E. coli* proteins [21].

## Anti-SmLEV1.3 polyclonal antibody production and western blot analysis

All SDS-PAGE separations were conducted as described previously [63]. Successful rSmLEV1.3 purification was confirmed by SDS-PAGE separation, followed by Coomassie Blue staining [64] and anti-His$_6$ western blot analysis. Predicted rSmLEV1.3 proteins bands were excised, and in-gel trypsin digestion followed by mass spectrometric analysis was conducted [63]. Subsequent sequence identification using Mascot (version2.2.1; Matrix Science) database search of the *S. mansoni* predicted proteins (genome version 7) confirmed the sole presence of rSmLEV1.3 (S2 Table).

Antiserum against rSmLEV1.3 was raised in rabbits by Lampire Biological Labs (USA) with rabbit pre-bleeds obtained before administration of rSmLEV1 protein. The anti-SmTSP-2 antibody was a kind gift from Professor Alex Loukas (James Cook University, Australia). For western blotting, 1.5 µg of parasite extract or 100 ng of each EV fraction was mixed with 2x SDS non-reducing sample buffer (0.143 M SDS + 0.125 M Tris (pH 6.8) + 20% v/v glycerol) and incubated for 5 min at 100˚C, prior to separation by 1D-SDS-PAGE and transfer to poly-vinylidene difluoride (PVDF) membrane as described previously [63]. Membrane blocking occurred for 2 h at room temperature in blocking buffer (PBS containing 0.2% fish skin gelatine 0.3% Tween 20). Membranes were then probed with anti-rSmLEV1.3 at a dilution of either 1:2,000 (extracts) or 1:20,000 (EVs), or with anti-SmTSP-2 (1:2,000 dilution) in blocking buffer overnight at 4˚C. The membrane was washed 3–5 times for 15 min in Blocking buffer before incubation with goat anti-rabbit IgG peroxidase-conjugated secondary antibody (1:10,000; Promega) in blocking buffer for 45 min. Blots were developed using SuperSignal West Pico PLUS (Thermo Scientific) according to the manufacturer's instructions. Western blots were then imaged in a UVITEC Alliance Q9.

## Immunohistochemistry (IHC) staining

Fixed, permeabilised cercariae were incubated in blocking solution (PBS containing 5% goat serum, 0.3% Triton X-100, 0.05% Tween 20) with gentle agitation for 2 h at room temperature and then further incubated with primary antibody (anti-rLEV1.3 or rabbit pre-bleed serum) diluted (1:50) in blocking solution overnight at 4˚C with gentle agitation. Samples were washed 3 times with PBSTx (PBS containing 0.3% Triton-X 100) for 10 min at room temperature with gentle agitation before being incubated in goat anti-rabbit Alexa Fluor 594 (Thermo-Fisher Scientific, 1:500) overnight at 4˚C. Samples were washed twice with PBSTx for 40 min at room temperature and finally with PBSTx containing DAPI (1 µg/mL final conc.) before being mounted and visualised on a SP8 Leica super resolution laser confocal microscope.

Lectin stained parasites were also fixed and permeabilised as described previously [24] and then directly incubated with Fluorescein labelled Peanut Agglutinin (PNA) (4 µg/mL) in PBSTx and incubated overnight at 4˚C. Parasites were then washed 3 times in 1 mL of PBSTx for 40 min at room temperature before being mounted and visualised.

## Human study population

The study cohort comprised of inhabitants of Musoli, a fishing community on Lake Victoria, Uganda. Descriptions of cohort selection, quantitative parasitology, and treatment regimens for this study can be found in a previous publication [25]. This report focused on 211 male members of the cohort who were under 50 years of age and who donated blood before and 9 weeks after treatment with one dose of 40 mg/kg PZQ. Infection intensity was expressed as eggs per gram (epg) of faeces, calculated from Kato-Katz thick smear counts on 3 stool samples collected on different days (2 smears per sample).

### Human anti-SmLEV1.3 serological analysis

Human IgE, $IgG_1$, and $IgG_4$ levels against recombinant rSmLEV1.3 were measured by ELISA as described previously [25]. The rSmLEV1.3 coating concentration was 6.25 μg/ml, as determined by anti-His$_6$ antibody (ThermoFisher) and coating inhibition assay [65]. Biotinylated mouse anti-human IgE (clone G7-26, Pharmingen), biotinylated mouse anti-human $IgG_1$ (clone G17-1, Pharmingen), or biotinylated mouse anti-human $IgG_4$ (clone G17-4, Pharmingen) were used at 0.5 μg/ml for detection of human anti-SmLEV1.3 antibodies. Detection was achieved by 1 h incubation with 1:3,000 HRP-conjugated streptavidin (Mast Group Ltd), followed by development in o-phenylenediamine substrate solution (Sigma). Development was stopped with 2 M sulphuric acid. Standard curves were generated with control immunoglobulins for IgE, $IgG_1$, and $IgG_4$ as previously described [38]. Plasma samples from 13 non-endemic donors were used in each assay as a non-infected control.

To account for the nonlinear relationship between infection intensity and age, individuals were divided into the following age groups: 7–9 yrs. (n = 39), 10–13 yrs. (n = 41), 14–23 yrs. (n = 42), 24–37 yrs. (n = 63), 38–50 yrs. (n = 26). Individuals were classified as a binomial variable: "Responder" and "Non-responder". $IgG_1$ and $IgG_4$ responders were individuals with a pre-treatment antibody titre greater than the mean + 3 standard deviations of the response by the plasma panel donated by the 13 individual non-endemic samples (NES). This was not suitable for potential IgE responders as the variation in the NES panel was too low to create a suitable threshold. Therefore, the mean + 3 standard deviations of the raw Optical Density (OD) was used to calculate the responder threshold. Variation in the proportion of anti-rSmLEV1.3 responders across the age groups of infected human individuals was tested for statistical significance with a $Chi^2$ test. Longitudinal statistical analysis of the same individuals (anti-rSmLEV1.3 $IgG_1$ pre-and post-treatment with PZQ) was analysed using a Wilcoxon signed-rank test. Statistical analysis of unmatched groups (EPG faeces in responders vs non-responders) was conducted using a Student's $t$-test post-hoc analysis (least significant difference).

### Mouse vaccination regime

Experimental C57BL/6 mice (n = 14) received the vaccine formulation consisting of 20 μg recombinant rSmLEV1.3, 10 μg ODN1826 Class B CpG Oligonucleotide (Invitrogen) and 100 μl Imject Alum (ThermoFisher) made up to 250 μl with PBS. Control mice (n = 14) received the same adjuvant formula without rSmLEV1.3. Mice were given a total of three immunisations 21 days apart, each of 250 μl via intra-peritoneal (IP) injection on alternating sides of the abdomen. Whole blood was collected via tail vein bleeds on day -1 (1-day pre-vaccination), day 49 (7 days post-vaccination), and day 84 (7 days post-parasite challenge) and via cardiac puncture upon sacrifice on day 126. For the preparation of serum, whole blood was left to coagulate at RT for 2 h, refrigerated o/n at 4˚C and then centrifuged at 10,000 x g for 10 min. Serum supernatant was removed and stored at -80˚C for further use. All mice were challenged 35 days after the final immunisation with 150 cercariae per mouse. Seven weeks later, adult parasites were perfused from mice as described [54]. Afterwards, male and female parasites were separated by incubation with a 0.25% solution of anaesthetic (ethyle-3-aminobenzoate methanesulfonate (Sigma Aldrich)) in culture media for 15 min before being counted and photographed for calculation of worm size using the pixel counting function in ImageJ [66]. Once perfused, small snips of liver were taken from each lobe of the infected animal, combined, weighed, and digested overnight in 5% KOH. Similarly, the small and large intestine were removed, flushed to remove faeces, weighed, and digested in 5% KOH overnight. Egg counts were then conducted to calculate the egg burden/g tissue. Once perfused, and snips taken for egg counts, the remaining liver was transferred to 10 ml of 10% buffered formalin

(for histological fixation). Liver samples were fixed at room temperature for 12 h. Histology sectioning and staining with Haematoxylin Eosin (H&E) was conducted at the University of Cambridge, Department of Pathology, UK. In brief, livers were embedded in paraffin wax and sectioned to create 5 μm slices prior to staining with H&E. Up to 10 individual granulomas per mouse were measured in diameter, the radius calculated as half the diameter, and the volume (V) calculated (assuming a spherical shape for each granuloma), using the formula:

$$V = \frac{4}{3}\pi r^3$$

### Murine anti-SmLEV1.3 serological analysis

rSmLEV1.3 was coated onto Immunlon 4HBX 96 well plates at a concentration of 6.25 μg/ml as before. Experimental mouse sera were used to probe rSmLEV1.3 at a dilution 1:100 in wash buffer (PBS containing 0.05% Tween-20) with 1% bovine serum albumin (BSA). HRP-conjugated anti-mouse $IgG_1$, anti-mouse $IgG_{2b}$ (Invitrogen) and anti-mouse IgE (ThermoFisher) were diluted 1:1,000 in wash buffer with 1% BSA. As a negative control, the secondary antibody alone was included on every plate. Development was achieved with 2,2′-azino-di-(3-ethylbenzthiazoline sulfonic acid) (ABTS) substrate solution (Sigma Aldrich); the reaction was stopped after 15 min with 1% sodium dodecyl sulphate (SDS) and the optical density (OD) read at 405 nm, using a Polarstar Omega Plate reader (BMG Labtech, Offenburg, Germany). Statistical analysis of unmatched groups (vaccinated vs controls) was conducted using a Student's *t*-test post-hoc analysis (least significant difference). Longitudinal statistical analysis of the same individuals (i.e., pre-, and post-vaccination) was analysed using a Wilcoxon signed-rank test.

### Supporting information

**S1 Fig. SmLEV1 homologues are present in other *Schistosoma* species, with high levels of sequence similarity.** (A) MUSCLE Alignment of the deduced full-length amino acid sequences of LEV1 homologues (S1 Table) visualised by JALview [67]. Conserved amino acid residues are shaded blue, residues with a positive BLOSUM62 score are shaded lilac, non-conserved residues are white. Exon numbers (relative to the full-length SmLEV1.1) are indicated above, and probable exon duplications (identified via tBLASTn) named Exon#.2. (B) Phylogenetic tree of deduced LEV1 homologues was inferred by using the Maximum Likelihood method based on the JTT matrix-based model [68] and drawn using MEGA7 (http://www.megasoftware.net/). The tree with the highest log likelihood (-2154.30) is shown. Node values indicate percentage of trees in which the associated taxa clustered. The tree is drawn to scale, with branch lengths measured in the number of substitutions per site.
(TIF)

**S2 Fig. *smlev1* and *sjlev1* demonstrate developmentally regulated expression.** (A) Analysis of data derived from a previous DNA microarray study [19] quantitatively shows a significant abundance of *smlev1* in cercaria compared to all other lifecycle stages, and a 10.7-fold higher expression in the male vs female. Contig_44 represents a 50 bp oligonucleotide spanning a region comprised of exon 10 and 11. The specific life-cycle stage is shown along the x-axis, mean normalised fluorescence units is shown along the y axis. Error bars represent standard error of the mean normalised fluorescence units. Significance compared to cercaria stage is indicated as: p<0.001 ***; p<0.005 **. (B) The *S. japonicum lev1* homologue (Sjp_0090520) exhibits developmentally regulated expression similar to *smlev1*. Data derived from a previous

*S. japonicum* microarray study by Gobert *et al* [20] shows the *sjlev1* homologue also exhibits developmentally regulated expression. Contig3607_451 (grey) represents a 60 bp oligonucleotide spanning exon 5 and 6 (equivalent to *smlev1* exon 8 and 9), whilst Contig3607_491 (black) represents a 60 bp oligonucleotide spanning exon 6 and 7 (equivalent to *smlev1* exon 9 and 10); the sequences detected by these two probes overlap each other by 20 base pairs. The specific life-cycle stage is shown along the x-axis, Average Cyanine 3-CTP intensity is shown along the y axis.
(TIF)

**S3 Fig. Production of recombinant SmLEV1.3.** rSmLEV1.3 production resulted in three protein bands <30 kDa in size visualise by SDS-PAGE and either stained by (A) Coomassie blue or transferred to PVDF and probed via (B) anti-His$_6$ antibody (used at 1:20,000). All bands contained peptides identified as SmLEV1.3 (Smp_074560.3) and no other SmLEV1 isoform, or *E. coli* proteins, when subjected to BLAST analysis against the GenBank database (p<0.05) (available at ncbi.nih.gov) (S2 Table). (C) Mapping peptides to the sequence (underlined) suggests that bands 2 and 3 represent successive N-terminal breakdown of rSmLEV1.3. M—protein standard, mass in kilo Daltons.
(TIF)

**S4 Fig. Presence of the EV marker Sm-TSP-2 in adult worm density fractions.** EVs from mixed-sex *S. mansoni* adult worms were isolated and subjected to density gradient separation as described, before probing with anti-SmTSP-2 Ab (1:2,000). Fractions containing SmTSP-2 (4–9) were combined and used for probing with anti-SmLEV1 pAb. Fraction numbers are indicated above the blot; the respective density ranges in g/cm$^3$ are shown below the plot. Secondary anti-rabbit HRP conjugated antibody was used at 1:10,000. M = protein standard, mass in kilo Daltons.
(TIF)

**S5 Fig. Immunohistochemistry staining of the cercarial head, reveals rabbit pre-bleed serum is reactive towards anterior peripheral extensions.** Permeabilised cercariae were stained with rabbit pre-bleed serum (1:50) followed by goat anti-rabbit Alexa Fluor 594 secondary antibody (Fisher Scientific, 1:500), then PBSTx containing DAPI (1 μg/mL). Cercariae were mounted and visualised on a SP8 Leica super resolution confocal microscope. Asterisk indicates staining of peripheral extensions also seen after staining with anti-SmLEV1.3 (Fig 6).
(TIF)

**S6 Fig. SmLEV1 elicits an age independent IgE response in an endemic human cohort.** Descriptions of cohort selection, quantitative parasitology, and treatment regimens for this study can be found in a previous publication [25]. rSmLEV1.3 was probed with biotinylated mouse anti-human IgE (Pharmingen). Detection was achieved with HRP-conjugated streptavidin (1:3,000 Mast Group Ltd), followed by development in o-phenylenediamine substrate solution (Sigma). (A) SmLEV1 IgE responders made up 11% of the Ugandan male cohort. The anti-SmLEV1.3 IgE response decreased with age and (B) no significant drop is seen in the anti-SmLEV1.3 IgE response after treatment. The number and percentage of responders are shown for each age group. The boxplot shows the median (line) with the upper and lower quartile (box) as well as the upper and lower observation within 1.5x interquartile range (IQR) (whiskers). Individual data points are also plotted.
(TIF)

**S7 Fig. Worms perfused from rSmLEV1.3 vaccinated, or control mice do not significantly differ in size.** Upon termination of the vaccine regime (7 weeks post cercarial challenge), adult

worms were perfused and anaesthetised for photographing. Images were processed for pixel counting in ImageJ [66]. No significant difference in worm size was seen between vaccinated (mean worm area per mouse: 1,934 pixels) and control animals (mean worm area per mouse: 2,081 pixels). The line and whiskers represent the mean with the standard deviation. Individual data points are also plotted.
(TIF)

**S8 Fig. Smp_306860 shows no significant variation in transcript abundance across 5 mammalian lifecycle stages.** Data from the previous *S. mansoni* DNA microarray [19] was mined to identify a suitable reference gene across cercariae, 24 h schistosomula, 3-day schistosomula, 7-week male- and female adult worms. The contig AF521086.1, mapping to Smp_306860 exhibited the lowest variation in abundance, with no statistically significant differences between any of the lifecycle stages. SmAT (alpha tubulin) was significantly higher in 7-week adult male worms than in cercariae ($p < 0.05$, +). The specific life-cycle stage is shown along the x-axis, mean normalised fluorescence units ± sem (standard error of the mean) were normalised to 7-week adult males (shown on the y axis) to enable the two contigs to be represented together.
(TIF)

**S1 Table. SmLEV1 homologues identified in other Schistosomatidae species.** The entire collection of genomes available at WormBase Parasite (available at parasite.wormbase.org) was interrogated for SmLEV1 homologues using the full-length (SmLEV1.1) protein sequence a search query. Initial BLASTp searches identified existing gene predictions in *Schistosoma* species. Secondary tBLASTn searches were then conducted to account for incomplete gene predictions and species with a draft genome. E value threshold was set to $1xE^{-5}$.
(XLSX)

**S2 Table. Unique SmLEV1 peptides identified by BLAST analysis.** Three proteins bands resulting from rSmLEV1.3 production (S3A Fig) were excised and subjected to mass spectrometric analysis. Sequences were identified using Mascot (version2.2.1; Matrix Science). BLAST analysis conducted against the GenBank database (available at ncbi.nih.gov) using *S. mansoni* genome version 7 revealed all bands contained peptides identified as SmLEV1.3 (Smp_074560.3) and no other SmLEV1 isoform, or *E. coli* proteins ($p < 0.05$).
(XLSX)

**S3 Table. Primers used for qRT-PCR.** Intron-spanning primers were designed to amplify Exon-1 (constant region) and Exon-2, Exon-5, and Exon-8 (variable regions) of the *smlev1* sequence. Primers were designed for Smp_306860 and compared with previously published *smat* primers for use as a housekeeping gene.
(XLSX)

## Acknowledgments

We thank current and past members of the Hoffmann laboratory and Ms Julie Hirst for contributing to schistosome lifecycle maintenance and vaccination experiments.

## Author Contributions

**Conceptualization:** Thomas A. Gasan, Iain W. Chalmers.

**Data curation:** Thomas A. Gasan.

**Formal analysis:** Thomas A. Gasan, Shona Wilson.

**Investigation:** Thomas A. Gasan, Marije E. Kuipers, Grisial H. Roberts, Gilda Padalino, Josephine E. Forde-Thomas, Jakub Wawrzyniak.

**Resources:** Shona Wilson, Edridah M. Tukahebwa.

**Supervision:** Karl F. Hoffmann, Iain W. Chalmers.

**Validation:** Karl F. Hoffmann, Iain W. Chalmers.

**Writing – original draft:** Thomas A. Gasan.

**Writing – review & editing:** Shona Wilson, Karl F. Hoffmann, Iain W. Chalmers.

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
