## [Decision Letter · Decision Letter 0]

19 Oct 2021

Dear Dr. Chalmers,

Thank you very much for submitting your manuscript "Schistosoma mansoni Larval Extracellular Vesicle protein1 (SmLEV1) is an immunogenic antigen found in EVs released from pre-acetabular glands of invading cercariae" for consideration at PLOS Neglected Tropical Diseases. As with all papers reviewed by the journal, your manuscript was reviewed by members of the editorial board and by several independent reviewers. The reviewers were highly complementary towards your paper and felt it was very complete, well performed and relevant. Based on the reviews, we are likely to accept this manuscript for publication, providing that you modify the manuscript according to the minor review recommendations. 

Sincerely,

John Pius Dalton, PhD

Associate Editor

Sergio Oliveira

Deputy Editor

Reviewer's Responses to Questions

**Key Review Criteria Required for Acceptance?**

**Methods**

-Are the objectives of the study clearly articulated with a clear testable hypothesis stated?

-Is the study design appropriate to address the stated objectives?

-Is the population clearly described and appropriate for the hypothesis being tested?

-Is the sample size sufficient to ensure adequate power to address the hypothesis being tested?

-Were correct statistical analysis used to support conclusions?

-Are there concerns about ethical or regulatory requirements being met?

Reviewer #1: Yes

Reviewer #2: No new analyses are required.

**Results**

-Does the analysis presented match the analysis plan?

-Are the results clearly and completely presented?

-Are the figures (Tables, Images) of sufficient quality for clarity?

Reviewer #1: Yes

Reviewer #2: Yes

**Conclusions**

-Are the conclusions supported by the data presented?

-Are the limitations of analysis clearly described?

-Do the authors discuss how these data can be helpful to advance our understanding of the topic under study?

-Is public health relevance addressed?

Reviewer #1: Yes

Reviewer #2: Yes

**Editorial and Data Presentation Modifications?**

Reviewer #1: (No Response)

Reviewer #2: (No Response)

**Summary and General Comments**

Reviewer #1: This is a really nicely written characterization of an abundant S. mansoni EV protein. I commend the group for their thorough and straightforward analysis and presentation of a considerable amount of work. Some comments:

134: Can the transcript abundance analysis by itself tell us anything about which transcripts (of the various splice variants) are present and in what proportions? 

146: Report whether the pattern of expression of exon 1 (as measured by qRT-PCR amplification) matches that shown from RNA-seq read analysis, in figure 2A?

159: The abundance of exon 2,5,8 transcripts in females and males seems (roughly) the same (fig 2B), yet the smlev1 RNA-seq data shows substantially higher numbers for males (Fig 2A). Can the authors explain this discordance?

269: The total tally (n) in fig 5A is 42; where does the number 46 come from?

598: Report how the cercariae were fixed and permeabilized?

Fig 6D: Any thoughts as to why the levels of IgE differ (significantly?) between the groups pre-vaccination? 

Fig 7D: Did the authors only measure the volume of 10 granulomas per group? If yes, doesn’t this undercut the veracity of their conclusion? Why so few? 

S1 Fig: Can the authors identify exon 5.2 in the S. mansoni genome?

S2 Fig: Why show us contig_44 here? 

S7 Fig: Consider reporting what area 2,000 pixels (say) actually represents.

S8 Fig: Say what SmAT is.

Reviewer #2: This is a well written and interesting manuscript that investigates for the first time an EV protein termed SmLEV1. The authors demonstrate that this is the most abundant protein in S. mansoni schistosomula EVs and that the protein is only found in the Schistosomatidae. The protein is found to be differentially spliced across the parasite life cycle and is localised to the pre-acetabular glands of cercariae. Interestingly Ugandan fisherman infected with the parasite display strong IgG1 responses to SmLEV1. The finding that vaccinated mice display reduced granuloma volumes is exciting and hints at potential therapeutic routes emerging from this work. The study seems to have been thoroughly conducted and the methods are well described. The Introduction provides an excellent grounding in work done to date, the results are well presented, and the discussion is appropriate and interesting. The data support the main conclusions of the paper. Overall, this paper will be of interest to those in the schistosome research community and those studying aspects host-parasite interactions and EVs in general. 

Minor points:

Author summary: I feel it should state somewhere in this section that schistosomes are human blood parasites.

I presume that somules were not stained with anti SmLEV1 antibodies? Although not essential this may have revealed the extent to which the protein is released during transformation and somule development over 72 h.

Could a different adjuvant have induced a stronger immune response? This could be discussed briefly in relation to the different adjuvants used in murine-based vaccination trials. 

There is some discussion (line 405 onwards) of the SMLEV1 protein potentially being present at the tegument of adult worms. Did the authors try to concentrate this worm fraction by stripping the tegument and running a greater amount of protein on a western blot? Immunolocalisation may also have helped here.

Minor, typographical etc:

Line 45: SmLEV1 rather than SmLEV-1?

Line 56: probably more correct to say “…start to penetrate the human skin…” as the biomolecules also facilitate the initial stage of penetration 

Line 60: change to “…many EV proteins…”

Line 72: I think S. mansoni should be in full as this is the first mention in the main text.

Line 78: ditto, probably ES needs defining here too.

Line 135: Hr is not an SI unit. Suggest change to ‘h’ here and throughout the manuscript.

Line 161: Legend Fig 2. It is important to say what the statistical comparisons are in relation to. In this case they are compared to the cercaria stage’. 

Line 219: the legend refers to AWA but the figure shows AW. Please correct.

Line 252: the legend refers to arrows as pink, but they are purple. Please correct.

Lines 318/319: probably worth indicating that these are mean values. 

Line 488: ‘sec’; suggest change to ‘s’ here and throughout the manuscript

Methods: Throughout the methods there is use of ‘minutes’ instead of ‘min’, hours instead of ‘h’ and also inconsistency of whether or not spaces are used between numbers and units. The authors are encouraged to tidy this up.

PLOS authors have the option to publish the peer review history of their article (what does this mean?). If published, this will include your full peer review and any attached files.

Reviewer #1: Yes: Patrick Skelly

Reviewer #2: No

Figure Files:

Data Requirements:

Reproducibility:

References

---

## [Editor Report · Decision Letter 1]

6 Nov 2021

Dear Dr. Chalmers,

We are pleased to inform you that your manuscript 'Schistosoma mansoni Larval Extracellular Vesicle protein1 (SmLEV1) is an immunogenic antigen found in EVs released from pre-acetabular glands of invading cercariae' has been provisionally accepted for publication in PLOS Neglected Tropical Diseases.

Best regards,

John Pius Dalton, PhD

Associate Editor

Sergio Oliveira

Deputy Editor

---

## [Editor Report · Acceptance letter]

15 Nov 2021

Dear Dr. Chalmers,

We are delighted to inform you that your manuscript, "Schistosoma mansoni Larval Extracellular Vesicle protein1 (SmLEV1) is an immunogenic antigen found in EVs released from pre-acetabular glands of invading cercariae," has been formally accepted for publication in PLOS Neglected Tropical Diseases.

Best regards,

Shaden Kamhawi

co-Editor-in-Chief

Paul Brindley

co-Editor-in-Chief
